# Navigating Trustworthiness of Deep Learning in $\Delta\Delta G$ prediction : Addressing Data Bias, Model Evaluation, and Interpretation

**Ruochi Zhang** [* 1 2]  **Ningning Chen** [* 1 3]  **Fengfeng Zhou** [4]  **Xin Gao** [1 5]

## Abstract

Artificial intelligence has emerged as an epicenter of global attention, given the rapid proliferation of cutting-edge AI tools. One promising avenue of application is the leveraging of deep learning methodologies to resolve complex biological conundrums. However, an essential question arises about the reliability and utility of deep learning models in the context of biosciences, where experimental data are often limited, especially in comparison to the vast data troves available in other domains. In this work, we focus on the task of identifying the change of binding affinity ($\Delta\Delta G$) induced by mutations in protein-protein interaction, exploring the impact of the data bias, the methods of model evaluation and interpretation. Surprisingly, we find that deep learning models may only learn the unintentional bias in the dataset instead of intrinsic principles, therefore proper data analysis and model evaluation should be applied not just focusing on improving the evaluation metrics. Our work provides a guideline to navigate the trustworthiness challenges in deep learning in bioscience and brings forth suggestions for future improvements.

## 1. Introduction

Protein-protein interactions (PPIs) play a crucial role in numerous fundamental biological processes, including DNA replication, signal transduction and immune response (So-leymani et al., 2022). Mutations in protein sequences can alter binding strength and thermodynamics of protein-protein interactions, which are typically assessed by binding affinity $\Delta G$. Understanding the change of binding affinity ($\Delta\Delta G$) caused by mutations is of utmost importance for protein engineering (Karanicolas & Kuhlman, 2009) and drug design (Macalino et al., 2018; Bruzzoni-Giovanelli et al., 2018). However, experimental methods for measuring binding affinity are often time-consuming and expensive. To address this challenge, computational methods show promise by utilizing accumulated data collected from the literature (Jankauskaitė et al., 2019). Recently, multiple machine-learning-based models have been proposed to predict the impact of mutations on binding affinity in PPIs. While most models aim to capture the geometric changes caused by mutations and require the 3D structure as input(Jiang et al., 2022; Liu et al., 2021; Rodrigues et al., 2019; Wang et al., 2020), MuPIPR (Zhou et al., 2020) solely utilizes the sequence information. Considering that the structure information of a complex is not always available and remains challenging to predict compared to individual protein chains (Evans et al., 2021). This work focuses on the sequence-based model with the goal of performing virtual mutation screening for protein engineering and drug design.

In the intricate domains of medicine and protein engineering, inaccuracies propagated by deep learning models can incur significant ramifications. Accordingly, we initiated our investigation with an exhaustive analysis of data, subsequently devising meticulous experiments to assess the model's credibility. Counter to our anticipations, we discovered that deep learning models may fail to sufficiently grasp the inherent association between the mutations and the alterations in binding affinity. Instead, they may inadvertently assimilate biases inscribed within the dataset. In pursuit of a deep learning model that truly addresses crucial real-world quandaries, we propose a systematic framework for scrutinizing model dependability and interpreting model predictions. We posit that our work holds the potential to aid researchers in refining their models, thereby enhancing their reliability for end-users.

Overall, our contributions are as follows:

---

[*]Equal contribution [1]Syneron Technology, Guangzhou, China [2]School of Artificial Intelligence, Jilin University, Changchun, China [3]ETH Zurich, Department of Biosystems Science and Engineering, Basel 4056, Switzerland [4]Key Laboratory of Symbolic Computation and Knowledge Engineering of Ministry of Education, Jilin University, Changchun, Jilin, China [5]Computational Bioscience Research Center, King Abdullah University of Science and Technology (KAUST), Thuwal, Saudi Arabia. Correspondence to: Fengfeng Zhou <FengfengZhou@gmail.com>, Xin Gao <xin.gao@kaust.edu.sa>.

*Accepted at the 1st Machine Learning for Life and Material Sciences Workshop at ICML 2024.* Copyright 2024 by the author(s).

- We addressed the concerns related to data distribution within the existing benchmark datasets employed for predicting mutation effects on protein-protein interactions. Our findings underscore that absent a meticulous evaluation, models might merely learn the bias innately embedded within the data.

- We formulated a comprehensive framework for assessing the reliability of models which consists of data analysis, model evaluation and interpretation.

- We introduced two sequence-based models that outperformed on the benchmark dataset by leveraging a more stringent evaluation procedure.

## 2. Experiments

Our objective is to establish a computational model capable of predicting the implications of mutations within protein-protein interactions, a crucial process in protein design and instrumental for advances in drug discovery. Nevertheless, prior to utilizing such predictions to steer protein design, it is imperative to assess the reliability of the model in question. Beginning with an in-depth analysis of the dataset, we devise a succession of experiments aimed at scrutinizing the proficiency of the black-box model. Additionally, we strive to elucidate the predictive mechanism of the model to gain a comprehensive understanding of its functionality.

### 2.1. Dataset and visualization

We collected the data from SKEMPI v2 (Jankauskaitė et al., 2019), which is the largest database providing the binding affinity of protein-protein interaction caused by mutations. Many previous works used this dataset as benchmark, but they used multiple subsets to test the performance and there is no one standard or well-defined benchmark. Here, we carefully cleaned the latest version of SKEMPI v2 and also analyzed the data to find if there is any bias in the dataset. This step is usually ignored by most of the model-developing papers as datasets are well-defined in cs fields, but it's essential in bioscience since there exists a significant imbalance in biological data (i.e. some proteins are well-studied, but others are not). Specifically, protein-protein interaction includes two particles which makes it more complicated. Therefore, we visualized the protein-protein interaction network into a graph and colored it by protein family collected from Pfam (Mistry et al., 2021). The results are shown in Figure 1. It's apparent that the dataset is highly imbalanced that some nodes are aggregated to form large clusters while others are segregated. This is a common problem in biological data (Chatterjee et al., 2023) and we will further interpret how it will impact the performance of deep learning models in the next section.

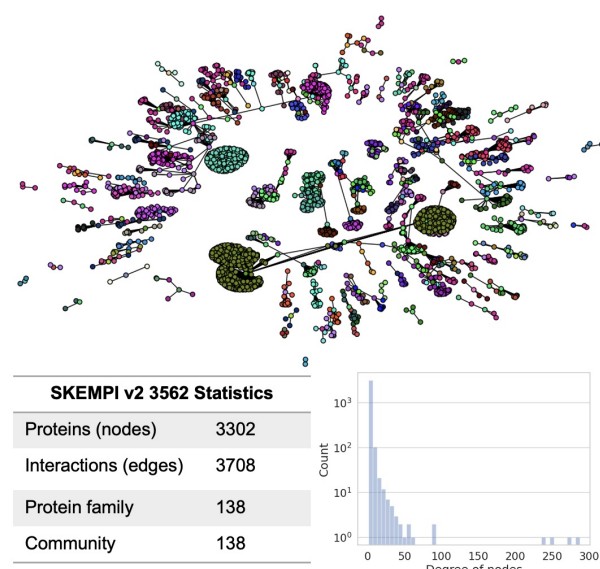

| SKEMPI v2 3562 Statistics | |
|---|---|
| Proteins (nodes) | 3302 |
| Interactions (edges) | 3708 |
| Protein family | 138 |
| Community | 138 |

*Figure 1.* **Cleaned SKEMPI v2 dataset visualization.** Graph plot of the 3562 doublet pairs of data, each node represents a protein sequence and each edge denotes an interaction between that protein pair. The nodes are colored by the protein family (Pfam). The table shows the basic statistics of the graph and the hist plot shows the node degree distribution of all protein sequences.

### 2.2. Models and set up

We evaluated three sequence-information-only models to investigate if deep learning models can really help to predict the effects of mutations on PPIs. To our knowledge, MuPIPR (Zhou et al., 2020) is the state-of-the-art model which is open-sourced focusing on this task. It first pre-trained a protein language model to contextualize the amino acid sequences and then trained Siamese residual recurrent convolutional neural network (RCNN) encoder with a multi-layer perceptron (MLP) regressor to predict the $\Delta\Delta G$ along with $\Delta G$ of wild-type pair and mutant pair of proteins in a multi-task manner. ESM model is a protein language model pre-trained on 200 million protein sequences and has shown superior performance on multiple downstream tasks (Rives et al., 2021). It is also proven that it can capture binding information in complexes. Therefore, we also evaluated if finetuning ESM model with an MLP regressor ($\text{ProtMut}_p$) can leverage the evolutionary information learned in the pretraining step to improve the performance on this task with limited data. In addition, we designed a model using cross attention ($\text{ProtMut}_c$) to capture the interaction information between protein pairs (Figure 6). We denote the model input as $I = \{(p_w, p_m)\}$, where $p_w$ and $p_m$ are the wild-type and mutant protein sequences respectively. Each pair contains two protein sequences $p_w = (s1_w, s2_w)$, $p_m = (s1_m, s2_m)$. The labels

are denoted as $O = \{\Delta\Delta G, \Delta G_w, \Delta G_m\}$.

## 2.3. Model evaluation

Deep learning models are developed to solve problems in real world settings, therefore, they should be carefully evaluated if it really performs well but not just overfit the training data before applying them to real world problems. In this section, we will propose a series of experiments to evaluate the performance of the models and try to interpret the model to understand the prediction mechanism.

**Dataset Split Methods** One important step in deep learning is determining how to split the dataset into training, validation and test sets. Protein sequences are consist of 20 different amino acids and vary in length. Based on these properties, we designed a dataset split strategy in addition to random split to evaluate the models in different settings and generalizability. We adopt a cluster-level split approach, we first calculated the pairwise sequence similarity of the $s1_w$ and $s2_w$ using a normalized Smith-Waterman algorithm and then clustered all the sequences based on the similarity by Hierarchical clustering algorithm. The total 381 unique sequences in the set $\{s1_w, s2_w\}$ are clustered into 209 clusters by the threshold of similarity 0.4. Each sequence pair $p_w$ is assigned a unique id based on the clustering results and the whole dataset is divided into five folds based on the clustering id.

**Metrics determination** The metrics to evaluate the model performance should be carefully chosen. It should align with the real world needs, which will be helpful for model users to choose the right model. In the task of estimating the change of binding affinity by mutations in PPIs, the possible application will be to optimize a protein to better bind a target or to introduce a mutation that might decrease the binding affinity between a protein and target to achieve a specific goal like increasing the specificity. Therefore, instead of the Pearson correlation and RMSE used by most of the models in this field, we used Spearman correlation as the metrics because the ranking between the mutations is more meaningful than predicting the accurate numbers of affinity. Additionally, the aggregate metric might only provide the "at a glance" performance of the model, but granular evaluations are needed (Burnell et al., 2023). In this task setting, the predictions inside one type of protein or protein family are more meaningful than the overall performance, i.e. the rankings between the different protein families might not be helpful.

**Held-out sanity check** Beside the traditional steps in deep learning, we also propose some approaches to sanity check that the model learns the basic knowledge but not something unexpected(e.g. data bias) like the phenomenon of "Clever Hans". Here, we propose two methods:

- Dataset features regression test: applying simple machine learning on the features of the datasets but not providing the sequence information. We implemented linear regression and gradient boosting regression on the features of sequence length, sequence degree in the graph and the number of mutations in each doublet input using the same 5-fold cross-validation.

- Held out one sequence in the protein sequence pair test: we only provide one sequence $\{s1_w, s1_m\}$ or $\{s2_w, s2_m\}$ to the model to evaluate if the model learns the interactions between the pairs.

## 3. Results

### 3.1. Model performance

As depicted in Figure 2, ProtMut$_p$ markedly surpasses other baseline models in both random split and cluster-level split scenarios. However, we observed a discernible decline in performance across all models upon implementing the cluster-level split strategy. For ProtMut$_c$ and ProtMut$_p$, the Spearman scores attained 0.75 and 0.81 respectively during the random split. However, following the application of the cluster-level split, these Spearman scores plummeted to 0.41 and 0.48 respectively. The MuPIPR model registered Spearman and Pearson correlations only marginally lower than ProtMut$_p$ under random split conditions, but these metrics dramatically fell below 0.2 upon the introduction of the cluster-level split. Given our prior analysis of the data, it is plausible that MuPIPR may be susceptible to biases inherent in the dataset. Furthermore, as illustrated in Figure 2, even in the absence of amino acid information, the gradient boosting regressor exhibits the capacity to construct a satisfactory model under random split conditions, achieving a Spearman score of 0.55 and a Pearson score of 0.69. However, the performance of the gradient boosting regressor significantly deteriorates when subjected to a cluster-level split, falling below 0.2. This suggests that the nonlinear combination of certain elementary features can also excel in a randomly partitioned test set. Nonetheless, such models fail to encapsulate the underlying principles of protein-protein interactions.

We executed a more intricate study on the test set under cluster-level split conditions, comprising a total of 50 groups of wild types and their corresponding mutants. In Figure 4, we showcase the distribution of Spearman scores of ProtMut$_p$ across these 50 groups. It can be discerned that although the model performs commendably in most groups, there exists a substantial variance across different groups. Furthermore, the Spearman correlation scores in some subsets dipped below 0.5 and even manifested negative correlations. Figure 7 furnishes a detailed portrayal of the model's predictions juxtaposed against the ground truth

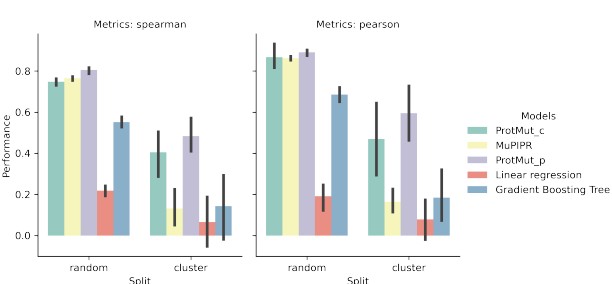

Figure 2. **Model performance on different dataset splitting strategy.** Five fold cross-validation results using random split and cluster-level split.

for each subset.

### 3.2. Model interpretation

We employed Principal Component Analysis (PCA) (Wold et al., 1987) to decrease the dimensionality of the representations acquired by ProtMut and MuPIPR, subsequently visualizing these in a two-dimensional space. As delineated in Figure 5, the X-axis and Y-axis denote Principal Component 1 and Principal Component 2 respectively. It is apparent that the Principal Component 1 of MuPIPR shares a potent correlation with sequence length, implying that the model may not be deciphering the underlying protein-protein interaction mechanism. Instead, it appears to be capturing dataset-specific characteristics, such as sequence length. In contrast, the embedding of ProtMut does not manifest a significant correlation with sequence length.

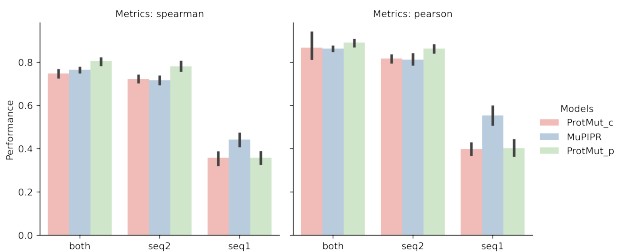

Figure 3. **Model performance on held-out inputs.** "Both" denotes the complete input, "seq1" and "seq2" denote that there is only one sequence in PPIs as input.

Figure 3 showcases the evaluation results of the held-out inputs sanity check. Intriguingly, when only seq2 (mutated sequences) information is supplied to ProtMut and MuPIPR, the performance remains largely unaltered. In the preceding Section 2.1, we dissected the SKEMPI dataset and identified it as a highly imbalanced dataset, replete with intricate relationships. For any given wild-type sequence,

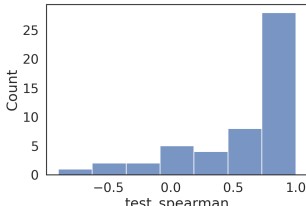

Figure 4. **Distribution of cluster-level spearman in the test set.**

there are typically multiple mutated sequences present in the dataset. This one-to-many relationship may cause the model to neglect information from the wild-type sequence (seq1). Moreover, mutated sequences might be linked to other wild types, leading to the same sequence being allocated to both the training and testing sets. This scenario allows the model to excel on the testing set by merely learning the characteristics of mutated sequences. This observation further accentuates the crucial need for conscientious test set partitioning when the dataset is highly imbalanced and teeming with complex relationships.

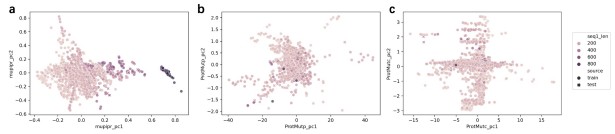

Figure 5. **Embeddings of models in the cluster-level split.** Embeddings of a. MuPIPR, b. ProtMut$_p$, c. ProtMut$_c$ colored by the sequence length are shown. The style of points denotes the data in the training set or the test set.

### 4. Conclusion and Discussion

This study explored the application of deep learning models in biosciences, specifically focusing on identifying changes in binding affinity($\Delta\Delta G$) caused by mutations in protein-protein interaction. We underscored the importance of proper data analysis and model evaluation as crucial determinants of the trustworthiness of such models. Our findings unveiled that deep learning models might be learning unintentional biases present in the dataset rather than the actual biological relationships, thereby questioning their practical utility. The results depicted that models like ProtMut performed significantly better than other baseline models under both random and cluster-level split conditions. However, the implementation of the cluster-level split strategy led to a marked decrease in performance across all models. This trend highlighted the sensitivity of these models to the partitioning strategy and the potential pitfalls of bias in the data. It also emphasized the necessity of rigorous model evaluation approaches that extend beyond mere metric im-

provement.

In conclusion, while deep learning offers immense potential in biosciences, its applications should be approached with caution. A thorough understanding of the underlying data and appropriate evaluation methods is essential to avoid misleading conclusions and to truly harness the power of deep learning in this field.

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

# A. Additional Tables and Figures

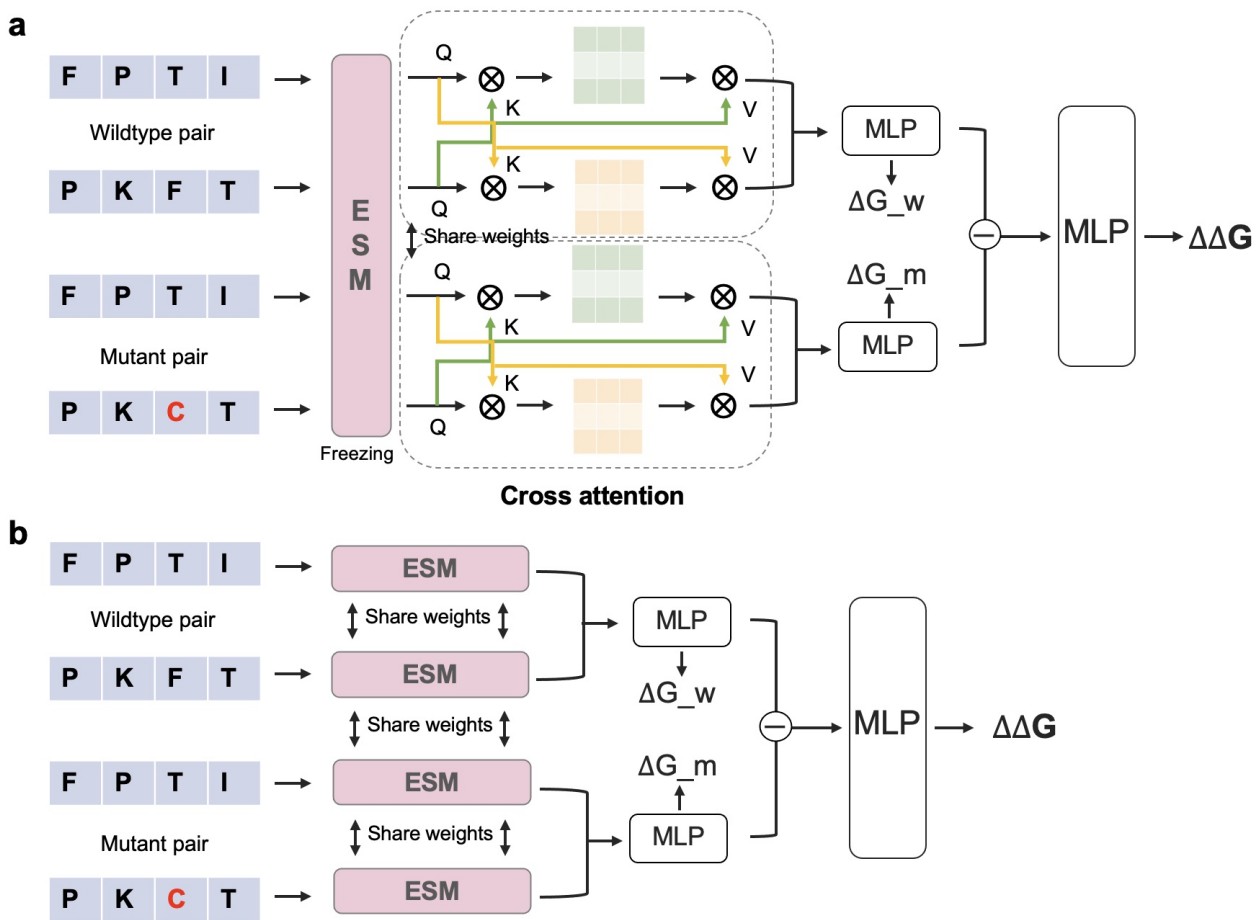

*Figure 6.* **Model architecture.** a. The architecture of ProtMut$_c$. b. The architecture of ProtMut$_p$

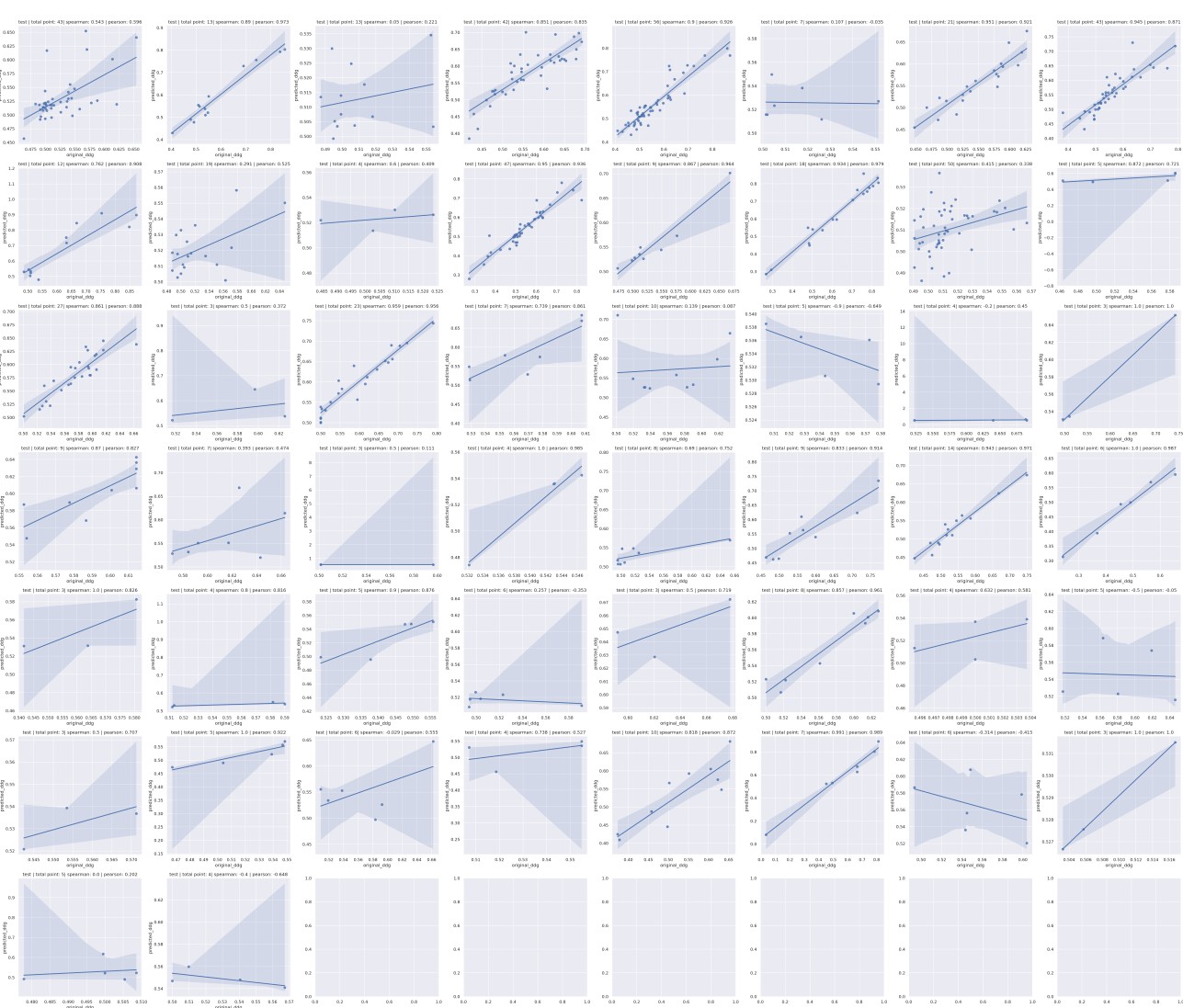

*Figure 7.* **Scatter plot of cluster-level results in the test set.** The Cluster-level metrics are shown in the title of each sub graph.