# OpenReview forum: "Navigating Trustworthiness of Deep Learning in ∆∆G prediction : Addressing Data Bias, Model Evaluation, and Interpretation"
_ICML.cc/2024/Workshop/ML4LMS — ML4LMS Poster_

### Official Review · Reviewer_6kBb · 2024-06-11
**On the importance of addressing composition and bias in biological datasets**

**Rating:** 9
**Confidence:** 4

**Review:**

The authors evaluate the impact of bias in the data (e.g. strong imbalance) on model predictions using a dataset of change in binding affinity upon protein mutations. They take a dataset of ~3.5k protein-protein interactions where either of the proteins is mutated and point out a large imbalance upon clustering it by families. The authors propose a data split strategy based on sequence similarity, a metric more suited to the task at hand and some sanity checks to detect "model cheating". They show that their proposed modeling approach convincingly outperforms the state of the art method and is less prone to the dataset bias.

Pros:
- The paper addresses a critical issue in applying deep learning to biology: dataset composition bias
- They point out critical drawbacks of other evaluated models, such as them only taking into account only the binding protein instead of both or their embeddings clustering by protein sequence length

Cons:

Quality: The results are convincingly demonstrated using relevant metrics. The paper is well written and organized.

Clarity: The motivation, methodology and results of the paper are clearly presented.

Originality:
The proposed modeling approach seems like a smaller improvement upon existing approaches but their handling of the dataset (splitting and eval metrics) is relevant and novel

Significance:
Addressing bias in biological datasets is of uttermost importance as lots of datasets are weighted towards certain well-studied proteins/sequences (e.g. GPCRs). This paper uses one such biased example and proposes a strategy on how to address the bias while still getting a useful model.

---

### Official Review · Reviewer_mS7x · 2024-06-12
**Framework to analyze reliability and effect of mutations on PPI models**

**Rating:** 6
**Confidence:** 5

**Review:**

The paper recaps existing models that can successfully predict PPI. They then test these methods with different dataset splits, accuracy metrics (Spearman's corr.) and held out data.

With these variations in training setting, they analyze the performance of the existing PPI models, and show that some models maintain performance across different settings while others do not. They posit that the models that are less robust are ones that are learning biases in the data.

Overall, I think this is a promising initial step to more thoroughly benchmarking PPI models, but it lacks significant contribution beyond analyze the variance in model performance. It would be nice to see the authors probe these models further to understand where the structured noise is prioritized architecturally and how we can mitigate that via implementation.

---

### Official Review · Reviewer_gYpZ · 2024-06-12
**Review of Submission 51**

**Rating:** 8
**Confidence:** 3

**Review:**

**Summary**
This paper aims to identify the change in binding affinity through mutual protein-protein interaction and provides a guideline to navigate trustworthiness in deep learning for bioscience.

**Strengths**
1. This paper is well-motivated and provides an in-depth analysis of proper data analysis and evaluation.
2. The visualization in terms of data distribution, model performance, and interpretation is well supported.

**Weaknesses**
1. It would be better if the authors further include the complexity analysis of deep learning models in terms of time and memory and discuss practical usage within current evaluations.
2. The legend of Figure 5 is too small, it is recommended that the authors present this in a horizontal format.